

# Monitoring changes in forestry and seasonal snow using surface albedo during 1982–2016 as an indicator

Terhikki Manninen[1], Tuula Aalto[1], Tiina Markkanen[1], Mikko Peltoniemi[2], Kristin Böttcher[3], Sari Metsämäki[3], Kati Anttila[1], Pentti Pirinen[1], Ali Nadir Arslan[1]

[1]Finnish Meteorological Institute, P.O. Box 503, FI-00101, Helsinki, Finland
[2] Natural Resources Institute Finland (LUKE) P.O. Box 2, FI-00791 Helsinki,Finland
3Finnish Environment Institute (SYKE), P.O. Box 140, FI-00251, Helsinki, Finland

*Correspondence to*: Terhikki Manninen (terhikki.manninen@fmi.fi)

**Abstract.** The surface albedo time series CLARA-A2 SAL was used to study trends in the snow melt start and end dates, the

10 melting season length and the albedo value preceding the melt onset in Finland during 1982-2016. The results were compared with corresponding snow melt timing calculated using the land ecosystem model JSBACH. In addition, the melt onset was compared with the greening-up timing based on MODIS data. Likewise the end of snow melt was compared with the melt-off day product by SYKE based on Fractional Snow Cover time-series provided by Copernicus CryoLand service and the FMI operational end of snow melt dates based on *in situ* measurements. It turned out that the albedo threshold 20 %

of the melting season dynamic variation corresponded well to the melt estimate of the permanent snow layer. The greening-up followed the albedo threshold 1% within 5 – 13 days, more rapidly in mountainous areas and more slowly on coastal areas. In two northern vegetation map areas a clear trend to earlier snow melt onset (0.5 – 0.6 days per year) and increasing melting season length (0.6 – 0.7 days per year) was observed. In the forested part of northern Finland a clear decreasing trend in albedo (0.2% - 0.3% per year in absolute albedo percentage) before the start of the melt onset was observed. The

increased stem volume explained the trend.

## 1. Introduction

Surface albedo is the fraction of incoming solar radiation reflected hemispherically by the surface. It is one of the essential climate variables (ECV). It serves as an indicator of climate change and change of the albedo will also affect the climate (GCOS, 2016). Regarding the net climate effect, both land carbon budget and properties of the land surface (e.g. albedo,

surface roughness) are globally significant (Davin 2010). The boreal forest zone is sensitive to changes in local and global climate (Parry et al., 2007) and the forest cover has a very significant influence on the northern hemisphere albedo being an important component of the northern hemisphere carbon budget (Bonan et al, 1992; Randerson et al., 2006). Change in albedo is an important mechanism by which forests modify climate in boreal regions, but the net effect due to simultaneous change in carbon sequestration is uncertain (Betts, 2000). If the length of the snow covered season decreases and snow melts

earlier in spring due to climate change, the albedo of forest areas decreases earlier in spring, which enhances the climate



change. On the other hand, if the northern forest edge moves further north due to the climate change, the winter and spring time albedo will show a notable decrease, which will enhance the climate change. In addition, changes in species, e.g. from deciduous (such as mountain forests typical of Arctic and Subarctic areas) to coniferous, would decrease the albedo of the snow covered season markedly. Previously it has been shown that the climatic and vegetational zones are equivalent in the

boreal forests (Solantie, 2005). Hence, a marked change in climate would result in change of climatic and vegetational zones.

Already changes in land management can have a significant effect on surface temperature, equal to that of land cover change (Luyssaert et al., 2014). The forest management changes albedo and carbon sequestration. Changes in forest management practices can markedly affect the boreal forest albedo through changes in leaf area index, biomass or canopy density,

especially in winter/spring when the forest floor is covered by snow. In Finland the forests are often managed, and the annual changes in forest biomass are monitored by in the National Inventories (Tomppo et al., 2011).

In future, the average temperature in Finland will rise more (Ruosteenoja et al., 2016a) and faster than the global average (Ruosteenoja et al., 2016b; Parry et al., 2007). Also the precipitation is estimated to increase, but more of it will be water

than snow in winter (Jylhä et al., 2012). On the average the changes will be larger in winter than in summer (Ruosteenoja, 2013; Ruosteenoja et al., 2016b). Warming will be fastest in Northern Finland (Ruosteenoja, 2013) and very low temperatures seem to become rarer. However, below freezing temperatures will persist in the northern parts of the country. It is anticipated that the snow covered period will become shorter especially in Southern Finland, while in Northern Finland the snow cover depth may actually increase due to increased snow precipitation. The Arctic warming has resulted in decline

of the Arctic Ocean ice cover (Vihma, 2014). There is evidence that such decline, together with changed heat and moisture budgets and increased snow cover in Eurasia, is connected to change towards atmospheric circulation patterns resembling the negative phase of the North Atlantic Oscillation and Arctic Oscillation, favouring cold winters in Europe and north-eastern Eurasia. According to observations, cold, snow-rich winters have become more common again in large parts of Europe since year 2005 (e.g. Cohen et al. 2010, 2012, Petoukhov and Semenov 2010, Osborn 2011).

Global climate models have challenges in representing the recent changes in the Arctic. The models may underestimate the observed negative Arctic Oscillation in response to sea ice and snow cover changes (Handorf, 2015). The strengthening and westward shift of the Siberian high-pressure system is too weak in the models as compared to reanalysis data (Dee et al., 2011). The models have deficits in the simulated changes in planetary wave propagation characteristics in response to sea ice

and snow cover changes. They have problems in the gravity wave response and upward vertical propagation (Vihma, 2014). They have not reproduced the observed winter cooling over large parts of Eurasia, and the sea ice and land surface components (or ecosystem models) of the climate models have difficulties in correct representation of the inter-annual variability of Arctic sea ice cover and terrestrial snow cover. Therefore, observations of snow and its annual cycle are needed





to discover the recent trends. Furthermore, if the land ecosystem model is applied in standalone mode to predict the evolution of snow cover, it is necessary to adjust the climate model output data with observational products.

EU Life+ MONIMET (http://www.monimet.fmi.fi) was an ambitious project spearheaded by scientists in Finland to increase turnover of climate data and to better understand climate change by mapping indicators of regional fluctuations that have an influence on the mitigation potential and vulnerability estimates of boreal forests and peatlands. The approach was based on a combination of different information sources describing phenology, $CO_2$ and $CH_4$ exchange, land cover, snow evolution and albedo (Böttcher et al., 2014; Bottcher et al., 2016). Satellite data sets provided by the MONIMET project are specifically used to derive indicators for the vegetation active period and the snow melt onset and melt-off day. MONIMET camera network was established to provide time series of vegetation (Linkosalmi et al., 2016) and snow observation (Arslan et al., 2017) and consists of about 30 cameras over Finland (Peltoniemi et al., 2018; Tanis et al., 2018).

In this work we study whether there were changes of significant scale in snow melt period and winter albedo during recent decades, 1982-2016, and compare the trends to changes in forest biomass. The surface albedo data record CLARA-A2 SAL (Anttila et al., 2016a, 2016b; Karlsson et al., 2017) was used to study trends in the snow melt start and end dates, the melting season length and the albedo value preceding the melt onset. We also use snow melt-off day derived from the Copernicus CryoLand Pan-European Fractional Snow Cover time series available for years 2001 – 2016 (Metsämäki et al., 2018). The snow melt end dates operationally produced at FMI from weather station observations are used as reference data. In addition, we use the land ecosystem model JSBACH (Raddatz et al., 2007; Reick et al., 2013) to derive snow melt onset and end days for the time range 1981 – 2011. The hourly climatic forcing for the land ecosystem model was produced with a regional climate model REMO (Jacob and Podzun, 1997; Jacob et al., 2001) that was constrained with ERA-interim weather (Dee et al., 2011) and further bias-corrected (Räisänen and Räty, 2013; Räty et al., 2014) with FMI gridded temperature and precipitation data (Aalto et al., 2013) to minimize the climate model inherent biases (Böttcher et al., 2016). The changes in the albedo values preceding the melting season are compared with trends in the stem volume available in the national forest inventory data sets per forests available at forest centres in order to evaluate, whether the changes in the forest explain the changes in albedo. An albedo model (Manninen and Stenberg, 2009) is used to link stem volumes / leaf area index and albedo.





## 2. Materials and methods

### 2.1. Data sets

#### 2.1.1. Vegetation zones

The digital vegetation zone map of Finland was provided by SYKE. It was reduced to 13 main regions (Figure 1, Table 1) by

removing the small islands from the map. Since the surface albedo product used in this study has the spatial resolution of just 0.25°, islands smaller than that would cause only mixed pixel problems. Even now the narrow coastal areas (Åland, Ostrobothnian coast) are prone to extra scatter due to mixed pixels. Hence, they were removed from the analysis. The vegetation zones stretch from hemi-boreal to northern boreal.

#### 2.1.2. Forest data

The forest inventory of Finland surveys forests of Finland based on uniform and dense sampling grid, and provides unbiased estimates of forest volume, and other forest associated variables. The results of the inventories have been estimated for 15 forest centres (Figure 1.) in inventory reports (Korhonen et al., 2000, 2000, 2001, 2013, 2017; Salminen 1993; Salminen and Salminen 1998; Tomppo et al., 1998, 2000, 2001, 2001, 2003, 2004). In this study we compiled time series of forest volume and area estimates of forests based on published inventory reports.

In the course of time, the sampling system of the national forest inventory has changed from regional to continuous sampling. In the regional sampling, the country was divided up to 15 forest centres. One or more regions were inventoried each year, while all regions were inventoried during the inventory period (5-7 years) after the full inventory period and when all forest centres were inventoried again in the next inventory period. More recently, in and since inventory no. 10, the whole

area of Finland has been inventoried within one year, but with lesser density. The full sampling density was reached within the inventory period, and forest resource estimates were again published. We assigned the forest volume estimates of newer inventories (10 and 11) to the mean years of the inventory periods. The northernmost forest centre, Lapland, further provides the estimates separately for the northern and southern parts in the most recent forest inventory.

The types and fraction of the forest area in each forest centre are shown in Table 2, as well as the average solar zenith angle value at melt onset time. The narrow coastal areas of Åland and Southern and Ostrobothnian coasts were removed from the analysis due to mixed pixel problems of the satellite based albedo product.

#### 2.1.3. Snow cover melt-off data

The snow cover melt-off day used operationally at Finnish Meteorological Institute is defined to be the first day after the

longest period of complete snow cover in a winter, when open areas have been continuously at least half covered by snow from the day of the beginning of permanent snow cover (Solantie et al. 1996; Kersalo and Pirinen, 2009). Complete





disappearance of snow cover in open areas is reached typically 10 days later (Solantie et al., 1996). The snow melt-off days were calculated using the *in situ* snow depth measurements and the results were interpolated in a 10 km x 10 km grid.

### 2.1.4. Surface albedo data

The data record used in this study (CLARA-A2 SAL) covers the years 1982-2015 and is based on homogenized AVHRR
data. It has been developed in the Satellite Application Facility project on Climate Monitoring, CM SAF, which is financially supported by European Organization for the Exploitation of Meteorological Satellites (EUMETSAT). The data record is described in more detail in Anttila et al. (2016a, 2016b, 2018) and Karlsson et al (2016 and 2017). The retrieved albedo is defined to wavelength range 0.25 -2.5 µm and the observations are averaged to 0.25° grid, which is also the resolution of the final product. The albedo values are given in the range 0 … 100 %. The annual mean surface albedo values
preceding the onset of snowmelt were determined for the 13 regions matching the vegetation zones of Finland and for the 16 forest centre areas of Finland. Also the mean starting and end dates of the snow melt and the length of the melting season were calculated for those regions (Section 2.2.1).

### 2.1.5. Fractional snow cover data

At-pixel Fractional Snow Cover (FSC, % of ground area covered by snow) is extracted from pre-Copernicus Cryoland snow
mapping service (Nagler et al., 2015),  which provides Pan-European FSC-maps starting from 2001 at 500m (0.05°) spatial resolution. The method applied in FSC retrieval is SCAmod (Metsämäki et al., 2005; 2012), complemented by some additional NDSI (Normalized difference Snow Index) rules for detecting the snow-free areas (Metsämäki et al. 2018). SCAmod detects FSC not only for non-forested areas but is also able to capture the under-canopy snow. This type of snow product featuring 'snow on ground', not just 'viewable snow' shows high FSC if forest floor (and under sub-canopy low
vegetation) is snow-covered even though from the satellite sensor's point-of-view the forest looks darker in terms of albedo. The gained accuracy (using FSC in-situ FSC observations at Finnish snow transects as reference) is 15-20 FSC %-units.

### 2.1.6. MODIS data

Daily Moderate Resolution Imaging Spectroradiometer (MODIS) data for the period 2001-2016 were utilized for the determination of the start of growth of deciduous canopy (section 2.2.4). The Terra MODIS level-1B data were retrieved
from the National Aeronautics and Space Administration (NASA)'s Level 1 and Atmosphere Archive and Distribution System (LAADS) and, starting from 2010 onwards, from the receiving station of the Finnish Meteorological Institute at Sodankylä. The data were processed to top-of-atmosphere (TOA) reflectances and projected into a geographic latitude/ longitude grid. The Normalized Difference Water Index (NDWI) was calculated from near-infrared (841–876 nm) and mid-infrared (1628–1652 nm) TOA-reflectances. Further details of the pre-processing are given in Böttcher et al. (2016).



### 2.2. Melting season start and end determination

#### 2.2.1. Start and end day of snow melt season from surface albedo data

Previously the end of the melting season of seasonal snow has been successfully estimated using standard deviation of weekly means of albedo data (Rinne et al., 2009). Now, the CLARA-A2 SAL pentad mean albedo values were studied by

5 fitting sigmoids matching the melting season using nonlinear regression similar to the method by Böttcher et al. 2014. For each pixel and year the pentads from end of January until end of August were used for this. The date of snow melt onset was taken to be the date at which the sigmoid reached 99 % of its variation range. Likewise the end of the snow melt season was defined to be the day at which the sigmoid reached 1 % of its variation range. The length of the melting season was then the difference between these two. The albedo value corresponding to the onset of melting was used as the representative albedo

value preceding the melting season (Anttila et al., 2018).

At first a rough sigmoid was carried out iteratively, the maximum number of iterations being five. After that, the albedo values lower than mid-sigmoid values were removed from the period preceding the melt season as temporary melt events. Likewise, the albedo values higher than mid-sigmoid values were removed from the period succeeding the melt season as

temporary snowfalls or cloud masking errors of the albedo product. Then the sigmoid was fitted anew using again a maximum of five iterations. In some areas right after the end of the snow melt new growth of vegetation started to increase the albedo level so soon that it affected the sigmoid fitting. Therefore the minimum albedo value starting from the mid-sigmoid was sought. A second order polynomial was fitted to the part of the albedo curve starting from that point. This regression polynomial was used to remove the growth season increase of the albedo but leave the pointwise variation of that

part of the original curve. The updated albedo curve was then used as the basis of a new sigmoid fit with maximum number of iterations being five. If the value of the coefficient of determination $R^2$ was smaller for the second sigmoid fit than for the first fit and $R^2$ was smaller than 0.99 for the first fit, the latter sigmoid fit was carried out anew with the maximum number of iterations being now increased to 15. Finally the sigmoid of the two rounds that had on the average smallest fit residuals was chosen.

In some cases the snow melt onset could not be determined, because it had started already before the first cloud free albedo pentad was available for the pixel and year in question. For the regional mean values of the melt onset and end dates and albedo at the melt onset day, only the pixels for which the melt onset day was available were included in the analysis.

#### 2.2.2. Start and end day of snow melt season from Ecosystem model

The JSBACH set-up used requires seven mutually consistent meteorological drivers: 2-m air temperature (Tair) and specific humidity, 10-m wind velocity, short-wave and long-wave radiation, potential short-wave radiation and precipitation (Pr) prescribed in hourly time resolution. For providing these drivers we used the regional climate model REMO (Jacob and



Podzun, 2007; Jacob et al., 2011) with an implementation of surface properties by Gao et al. (2015). The modelling domain for REMO was Fennoscandia, with a grid resolution of 0.167°, corresponding approximately to 18 km. Lateral boundary data was taken from ERA-Interim, a global atmospheric re-analysis produced by the European Centre for Medium- Range Weather Forecasts (ECMWF) (Dee et al., 2011). The forward run covered the period from 1979 to 2011 and it was preceded

by a 10-yearspin-up to equilibrate soil moisture and temperature.

Model-specific biases are inherent for regional climate models (Christensen et al., 2008; Teutschbein & Siebert, 2012) and it is known that REMO exhibits too cold winter temperatures in an Eastern European area that covers most of Finland (Pietikäinen et al., 2012; Gao et al., 2015). The modelled summer and autumn temperatures tend to exceed observations.

Moreover, REMO overestimates precipitation in northern Europe throughout the year. To account for the biases, we adjusted air temperature and precipitation against gridded homogenized weather data for 1980–2011 provided by the Finnish Meteorological Institute (FMI) (Aalto et al., 2013) using a quantile–quantile type bias correction algorithm for daily mean temperature (Räisänen and Räty, 2013), while daily cumulative precipitation was adjusted using parametric quantile mapping (Räty et al., 2014). Finally, the daily corrections were applied to the hourly modelled air temperature and

precipitation values.

JSBACH land surface model (Raddatz et al., 2007; Reick et al., 2013) resolves land surface physical and biogeochemical processes involved in surface energy balance as well as water and carbon balances within soil and vegetation. In addition to its principal use to serve as a land surface boundary for the Max Planck Institute for Meteorology Earth System Model

(MPI–ESM) (Stevens et al., 2013), JSBACH can be driven with prescribed weather data. In this work JSBACH was driven using the bias corrected hourly REMO data for the years 1980–2011 with a spin-up period of 30 years before the forward run to equilibrate soil water and soil temperature. The spatial domain of JSBACH is also Fennoscandia in a resolution of 0.167°, but in the present study only Finnish territory is investigated.

In JSBACH surface grid cells are divided into fractions of four most prevalent plant functional types (PFT) that are characterised with properties such as maximum leaf area, phenology type, growth rate, shedding rate and photosynthesis parameters. In addition to PFT fractions, each grid cell is characterised with a maximum fraction of the land area that is hospitable to plants.

Snowfall fraction of the total precipitation is given as follows:

$$
Pr\_sn = \begin{cases} 0, & Tair > 3.3°C \\ Pr * (3.3° - Tair)/4.4°, & -1.1°C < Tair < 3.3°C \\ Pr, & Tair < -1.1°C \end{cases} \tag{1}
$$



Pr_sn is further distributed to surface and canopy reservoirs with the constant fraction of 0.25 to be intercepted by the canopy. In addition to the snowfall fraction the fate of the canopy reservoir is constrained by sublimation and unloadings due to melting that is temperature regulated and due the wind-blow (Roesch et al., 2001). Furthermore the accumulation of snow is limited by leaf area index. At the ground the snow budget composes of the excess snowfall fraction after canopy interception, sublimation and melting.

The snow depth was used as an indicator of the melting season start and end. During early spring the changes in model albedo are very strongly dominated by changes in snow cover, since the changes in the leaf area of the coniferous forest are minor and the bud burst of broadleaved forest occurs later in spring. The modelled melting season timing and duration were studied by fitting a sigmoid to each grid cell of the yearly spring time snow depth data. The fit was done in two phases: first sigmoid was fitted between days of year 30 and 200 to include days of yearly maximum and minimum snow cover. As modelled snow cover did not typically show a plateau before the start of the melt period, but tended to increase monotonously until the start of the melting, values lower than the first fit were rejected from the part of the time-series prior to the point where sigmoid reaches its half of the range value. If fewer than 15 data points were left after the rejection, the grid cell in that year was rejected. Otherwise sigmoid was fitted to the remaining data series. After the second fit a sanity check was made to reject the fittings implying either too small or too large difference between the sigmoid parameter values representing the snow cover levels before and after the melting. Because the snow melt starts rather gradually and ends abruptly in the modelled snow depth data, asymmetric criteria of 98% and 8% of the sigmoid variation range was used for the start and end dates of the melting period, respectively. Finally, for each vegetation zone yearly regional means of start and end days were calculated.

### 2.2.3. Snow melt-off day from Fractional snow cover data

Snow melt-off day from Fractional Snow Cover maps is based in the detection of FSC time series. The detection of snow melt-off day from FSC provides the first day with snow-free terrain, however ignoring short (a few days) intervening snow-free periods within a long snow season (Metsämäki et al. 2018). The melt-off day from the FSC time series is identified as a beginning of a snow-free period (FSC = 0%) after a period of snow observations (FSC > 0%). A pixel is assigned the status "melted" after a period of at least six subsequent snow-free days is found (however with possible intervening cloudy observations), if the number of the snow-free days represents 80% of all cloud-free observations after the first day of the snow-free period in concern. The first day of this period is accepted as a candidate for the melt-off day. After the candidate day, a "new snow period" is allowed, if at least three subsequent snow days are observed. Then a new search for melt-off day is launched the same way as before.



### 2.2.4. Starting of growth season of deciduous canopy from MODIS data

The start of the vegetation active period of deciduous vegetation in Finland, usually referred to as greening-up, was determined from NDWI using the method described in Böttcher et al. (2016). The NDWI was utilized because the detection of the greening-up in boreal areas from the Normalized Difference Vegetation Index (NDVI) can be affected by melting of snow, whereas the NDWI allows a differentiation between snow melt and greening-up (Delbart et al., 2006). The start of the vegetation active period showed good correspondence with the date of birch bud break (RMSE of one week) at Finnish phenological sites when the corresponding MODIS pixels are dominated by deciduous forest. The data set for the period 2001−2016 has a spatial resolution of $0.5 \times 0.5$ degrees.

### 2.2.5. Albedo model calculations

The effect of snow cover on surface albedo of coniferous forests can be estimated using an albedo model based on photon recollision probability (Manninen and Stenberg, 2009). The forest parameters required are the leaf area index (LAI), the clumping factor and the single scattering albedo of needles. In addition, the sun zenith angle value is needed. The average value of the sun zenith angle of the forest centres at the start day of snow melt was used in the calculations (Table 2). The snow cover of the forest floor is described using its albedo. The forest inventory does not contain LAI information and a strong relationship between the stem volume and LAI can't be expected at high spatial resolution. Yet, in larger areas one could assume that LAI of a forest is statistically related to the stem volume per area $V$ according to LAI $= k\, V^{2/3}$. When comparing the albedo simulations to the satellite measurements in a larger area, one has to take into account also the fraction of forested area. The albedo was simulated in the forest centre areas for the years matching the national forest inventory during $1963 - 2011$ varying the coefficient $k$ in the range $0.1\ldots1.5$. The optimal value of $k$ was defined to be the value corresponding to smallest difference between the simulated and satellite based albedo values. It turned out that there was a relatively strong relationship between the optimal value of $k$ and the fraction of the forest area in the boreal forest zone (Figure 2). The area left outside the regression (Northern Lapland) is dominated by deciduous species (mountain birch), hence its LAI value in winter is not related to the stem volume similarly as for the coniferous species.

## 3. Results

### 3.1. Melting season of snow

#### 3.1.1. Snow melt onset

The onset of melting was determined using the sigmoids determined both from the satellite based albedo data CLARA-A2 SAL pentads and the daily JSBACH model calculations of the snow depth. The comparison of these melting season start dates is shown in Figure 3. The overall agreement is good, and some of the scatter can be attributed to using a different variable for the data (albedo) and the model (snow depth). Snow depth describes the whole snowpack, whereas the albedo



describes only the topmost layer of it. The surface albedo product showed a negative trend (about half a day per year) in two areas: Northern Karelia – Kainuu and North Ostrobothnia (Table 3, Figure 4). The annual variation is so large that it easily masks a long term trend. This was even more evident in the land ecosystem model calculation results, where a low frequency variation dominated the time window so that no area showed a marked trend even in the five year moving average curve.

The results, however, confirm the intuitive impression that the snow melt starts earlier than it used to do in the past, as the two distinct aerial trends showed.

### 3.1.2. End of snow melt

The end of snow melt was determined using the albedo data, the snow depth value of the land ecosystem model and the snow melt-off day derived from the satellite based fractional snow cover estimate for each area of the vegetation zone map.

The results were compared with the FMI snow melt maps provided operationally by the climate service of FMI (Solantie et al., 1996; Figure 5). The overall agreement is good for the melt-off day derived from MODIS-based FSC and the FMI operational melt-off-day derived using *in situ* data taking into account that the satellite based melt-off date refers to completely snow-free case, whereas the *in situ* based melt-off date refers to the end of the permanent snow cover, when half of open areas still have snow (Figure 5). The 1 % threshold for the albedo produced a clearly later end of melt day than the *in*

*situ* based snow melt-off day and model results. This is understandable, because the FMI operational melt-off day is not yet completely snow-free and the albedo value is very sensitive to even small amounts of snow. It turned out that the day at which the albedo based sigmoid reached 20% of its variation range matched well the official end date of snow melt and the value 15% matched well the FSC based melt-off day. The difference between dates corresponding to the 20% and 1% threshold values of the albedo based sigmoids is of the order of 13 days, which is in line with the 10 days difference between

open areas to be half or completely snow free (Solantie et al., 1996). The land ecosystem model results coincide well with the operational end of permanent snow cover. The reason for the end of melting season based on snow depth of the land ecosystem model to match the time, when half of open areas are snow-free is probably related to the snow-off bias inherited from the climate model surface parametrizations. It has been shown that e.g. in ECHAM5 the snowmelt occurs at too low temperatures (Räisänen et al., 2014). Consequently, too much of the surface net radiation is consumed in melting snow and

too little in heating the air.

No systematic trend was observed for the end of snow melt day for albedo or land ecosystem model data. It seems that the onset of snowmelt depends more clearly on the climate and the end of snowmelt is more sensitive to the prevailing weather conditions at that time.

The aerial advance of spring is demonstrated by the FMI operational melt-off map, the surface albedo 20 % threshold map, the FSC melt-off map and the greening-up map in years 2006 and 2012 (Figure 6). The general agreement is good, and the southeastern – northwestern zones of equal timing are obvious.





### 3.1.3. Length of melting season of snow

The length of the melting season was determined as the difference between the dates corresponding to the end of the snow melt and the onset of the snow melt. The 99 % and 1 % threshold values were used for the albedo sigmoids. A clear trend of increasing melting period length was observed in the areas, Northern Karelia – Kainuu and North Ostrobothnia, where the

snow melt starts earlier than before (Table 4, Figure 7).

### 3.2.  Starting of growth season of deciduous canopy

The start of the vegetation active period of deciduous vegetation based on MODIS NDWI values turned out to almost coincide with timing of the 1% threshold value of the surface albedo, being only slightly later (Figure 5). This is probably due to the large fraction of deciduous species being in open areas and forest floor, which start growing only after the soil has

become completely snow free and even small amounts of remaining snow have a large effect on the albedo. The 16 year average time between the date, when the SAL value reached 1% of its dynamic range during the melting season, and the date of the greening-up varied in the vegetation zones between 5 and 13 days. The largest difference took place on the coast and the shortest in mountain areas, where greening of mountain birch can occur even before snow melt (Shutova et al., 2006). In addition, trees in the northern boreal region have a lower temperature requirement for budburst than in the south and day

length is already longer in the north at the time of budburst. Therefore, the intensity of response to temperature increases from south to north and observed trend to earlier budburst was stronger in the northern boreal zone compared to the southern boreal zone for the period 1997-2006 (Pudas et al., 2008).

### 3.3.  Albedo before onset of snow melt

The surface albedo at the melt onset shows a decreasing trend in several northern regions based on vegetation zones (Table

5, Figure 8). This can be explained by the increase in stem volume, as the albedo model results agree with the observed albedo trends of the forest centre based regions (Table 6, Figure 9). As expected, no clear albedo trend after the snow melt was observed. The reason is that the difference between the canopy and forest floor albedo is not significant, whereas the snow and canopy albedo values differ markedly. Hence, the accumulating stem volume in the northern part of Finland causes darkening of the winter/spring time albedo, but does not have a significant effect on the albedo value right after the snow

melt.

### 4.  Discussion

The surface albedo is determined by the quantity and quality of land vegetation, the seasonal cycle of leaf development, and the existence of snow cover. The snow conditions also depend on the vegetation cover. Seasonal dynamics of snow and vegetation are driven by temperature and precipitation as form of snow or water. The air temperatures in the arctic and boreal

regions have increased and Arctic sea ice cover has reduced, and the influences can be seen in Finnish temperature and





precipitation records. The trends are not self-evident, although as a whole the arctic/boreal region is changing towards warmer climate. The large variation in the climate can be seen in both observed and modelled snow melt onset. The observed trends are clearly negative only in Northern Carelia – Kainuu and North Ostrobothnia. The year-to-year variation in the modelled onset is not as large as in the observed onsets but the multi-year variation is similar, model acting as a smoother of

5    observations. In many cases, both positive and negative tendencies appear during the study period, most significantly in around year 2000, when the snow melt showed a transition from earlier onset to later onset in Southwestern Finland, Lake District and Ostrobothnia. Also a previous analysis using 35 years data set from passive microwave radiometer snow clearance dataset, Metsämäki et al. (2018) concluded that snow melt-off at northern latitudes in Europe has advanced 3-5 days per decade for boreal forests and tundra. The MODIS FSC-based snow melt-off maps gave a similar signal.

The trends in end of snow melt are insignificant everywhere. However, combined with negative snow melt onset trend in Northern Carelia – Kainuu and North Ostrobothnia, we obtain an increase of length of snow melt period in those regions. Generally, in northern latitudes, many studies have shown persistent negative trends in snow melt and positive trends in early spring vegetation activity (Myneni et al., 1997; Pulliainen et al., 2017). A recent study indicated increase in the length of

snow melt period (Contosta, 2017), which could have both negative and positive impacts to e.g. soil water availability, nitrogen cycling and ecosystem carbon and energy balance and thus induce complex perturbation of land surface – climate feedbacks.

The forest growth in Finland can be linked in forest centre resolution to the sum of effective temperatures, the frost sum and

the mean snow depth during the winter in a very robust way (Solantie 2015). The snow cover affects directly the winter greatest soil frost depth. According to Solantie (2015) most of the increase in the growth-rate in forest stands in the southern half of Finland during 1961-2010 is caused by a change in climate. The snow cover affects the roots of the trees via the soil frost. In regions of deep soil frost the roots tend to concentrate in the surface layer, otherwise deeper down. Changes in snow cover are hence expected to affect also the amount of wind throw. In warming climate the roots will concentrate deeper

down and then in summer the canopy will tolerate higher wind speeds than before in otherwise similar conditions. On the contrary, late autumn, winter and early spring storms will be more dangerous for the canopy, if the soil is not frozen as it used to be in the past (Peltola et al., 1999). The traffictability of the forested terrain may reduce with earlier snow melt (Kellomäki et al., 2010).

The timber resources in Finnish forests have grown since the 1950's more than has been removed by logging (National Forest Inventory, 2013). In northern Finland the share of stout timber has traditionally been high, so that the increase comes mainly from small-diameter timber. While carbon storage in forests has increased Finland has darkened, which shows in the negative trend of albedo preceding snow melt season. The effect of canopy on snow covered terrain albedo is largest for small LAI values (Jääskeläinen and Manninen, 2018), hence it is understandable that the marked decreasing trend of albedo



before snow melt season is observed in northern Finland. In southern Finland the share of stout timber has more than quadrupled since the 1950's, yet no trend in pre-melt albedo during 1982-2016 is observed there. Hence, the idea to increase carbon sinks in forests by increasing the timber resources of existing forests by causing trees to grow more densely and become stouter (Pihlainen et al., 2014; Baul et al., 2017a, 2017b; Pingoud et al., 2018) suggests a possibility to combat

climate change in regions of already medium size canopy without the negative albedo feedback.

## 5.   Conclusions

The surface albedo turned out to be a suitable indicator of the melt onset and end of melting season. Overall agreement was found with corresponding dates determined using the JSBACH land ecosystem model snow depth data. The operationally provided end of permanent snow cover date based on *in situ* data corresponded to about the day when that surface albedo

value reached the lower 20% of its dynamic range during the melting season. The end of snow melt determined using satellite based fractional snow cover data and JSBACH land ecosystem model snow depth data agreed also well with the corresponding operationally produced date. The day, when the surface albedo reached the lower 1 % of its dynamic range is well in line with the day corresponding to completely snow free open areas. In addition, the greening-up day followed on the average 5 – 13 days later.

According to the albedo data, the melt onset day has advanced by about 17 and 21 days, i.e. 0.5 and 0.6 days per year, respectively,  in two regions in northern Finland (North Ostrobothnia and Northern Karelia - Kainuu), respectively. Likewise in those areas the length of melting season has increase by 23-24 days, i.e. 0.7 days per year. The albedo data also shows darkening albedo values (0.2% - 0.3% per year in absolute albedo percentage) preceding the snow melt onset in North

Ostrobothnia, Northern Karelia, Kainuu and Lapland, especially in Southern Lapland, which is coniferous species dominated. This albedo decrease could be explained by the observed increase in stem volume in those areas.

**Data availability**

The    surface    albedo    data    is    available    at    the    web    user    interface    of    CM    SAF    project:

https://wui.cmsaf.eu/safira/action/viewProduktSearch;jsessionid=5137E8E5FEF904478BBC386333FD8EB0.ku_2. The data on the vegetation active period (greening-up of deciduous vegetation) for the period 2001-2016 data can be viewed online in a web map application (http://syke.maps.arcgis.com) is and are available for download at the Open data service of SYKE (http://www.syke.fi/en-US/Open_information/Spatial_datasets#P). The original FSC data is available from the Cryoland portal (http://neso1.cryoland.enveo.at/cryoclient/). Forest inventory data is not publicly available, but aggregated values are

(http://kartta.luke.fi/index-en.html).





**Author contributions**

TeM performed the albedo data analysis, comparison with other data sets and albedo modelling. TiM carried out the ecosystem model calculations and analysis of those results. TA planned and led the climate related work and interpreted the simulation results. MP provided the forest inventory data, KB provided the greening-up data, SM provided the FSC-based yearly melt-off day data, KA provided the albedo data and PP provided the FMI snow melt data. AN coordinated the Monimet project and everybody participated in the scientific discussions. All authors participated in the writing of the paper.

**Acknowledgements**

The authors are grateful for EUMETSAT for financial support to generate the CLARA-A2 SAL time series in the CM SAF project. Life+ project Monimet (Grant agreement LIFE12 ENV/FI000409). The authors are grateful to Mr. Achim Drebs for discussions related to snow melt. The authors wish to thank also Ms. Maria Holmberg and Ms. Katri Rankinen for their co-operation at various phases of the project and for providing the vegetation zone map. We thank the Academy of Finland Center of Excellence (307331), OPTICA (295874) and CARB-ARC (285630) for support.

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



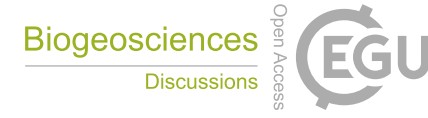

**Table 1. The regions based on the vegetation zones of Finland.**

| Region | Name | Type | Number of SAL pixels |
|---|---|---|---|
| 1 | Åland | Hemiboreal | 3 |
| 2 | Oak zone | Hemiboreal | 19 |
| 3 | Southwestern Finland | Southern boreal | 127 |
| 4 | Southern Ostrobothnia | Southern boreal | 19 |
| 5 | Lake district | Southern boreal | 224 |
| 6 | Northern Karelia – Kainuu | Middle boreal | 112 |
| 7 | Ostrobothnia | Middle boreal | 212 |
| 8 | Southwestern Lapland | Middle boreal | 23 |
| 9 | Kuusamo district | Northern boreal | 47 |
| 10 | North Ostrobothnia | Northern boreal | 130 |
| 11 | Northwestern Fjeld Lapland | Northern boreal | 20 |
| 12 | Forest Lapland | Northern boreal | 86 |
| 13 | Northern Fjeld Lapland | Northern boreal | 30 |





**Table 2. The regions based on the forest centres of Finland.**

| Region | Name | Mean solar zenith angle at onset of snow melt | Mean fraction of forested area [%] | Type | Number of SAL pixels |
|---|---|---|---|---|---|
| 1 | Åland | 64.9° | 59 | Hemiboreal | 40 |
| 2 | Southern and Ostrobothnian coast | 63.5° | 62 | Hemiboreal southern boreal | 63 |
| 3 | Southwestern Finland | 63.7° | 62 | Hemiboreal - southern boreal | 76 |
| 4 | Häme – Uusimaa | 62.4° | 67 | Southern boreal | 52 |
| 5 | Southeastern Finland | 63.3° | 74 | Southern boreal | 50 |
| 6 | Pirkanmaa | 62.9° | 75 | Southern boreal | 48 |
| 7 | Ostrobothnia | 64.6° | 70 | Middle boreal | 57 |
| 8 | South Ostrobothnia | | 71 | Middle boreal | 71 |
| 9 | Central Finland | 61.3° | 85 | Middle boreal | 64 |
| 10 | Etelä-Savo | 62.2° | 85 | Southern boreal | 60 |
| 11 | Pohjois-Savo | 60.4° | 81 | Middle boreal | 66 |
| 12 | North Karelia | 60.2° | 84 | Middle boreal | 68 |
| 13 | North Ostrobothnia | 60.3° | 77 | Middle – Northern boreal | 145 |
| 14 | Kainuu | 59.5° | 87 | Middle boreal | 77 |
| 15 | Southern Lapland | 59.0° | 81 | Northern boreal | 220 |
| 16 | Northern Lapland | 57.8° | 51 | Northern boreal | 103 |




**Table 3. Observed melt onset trends during 1982 – 2015 from the CLARA-A2 SAL data of regions based on vegetation zones. The slope and coefficient of determination values are shown both for annual data and moving averages of five years.**

| Region | Annual value | | Moving average of five years | |
|---|---|---|---|---|
| | Slope (Days per year) | $R^2$ | Slope (Days per year) | $R^2$ |
| Northern Fjeld Lapland | 0.21 | 0.024 | 0.13 | 0.041 |
| Forest Lapland | -0.003 | 0.0000067 | -0.07 | 0.02 |
| Northwestern Fjeld Lapland | 0.10 | 0.0076 | 0.06 | 0.014 |
| North Ostrobothnia | -0.26 | 0.071 | -0.33 | 0.33 |
| Kuusamo district | -0.21 | 0.047 | -0.33 | 0.34 |
| **Southwestern Lapland** | **-0.47** | **0.17** | **-0.50** | **0.78** |
| Ostrobothnia | -0.35 | 0.083 | -0.43 | 0.41 |
| **Northern Karelia - Kainuu** | **-0.53** | **0.19** | **-0.63** | **0.81** |
| Lake district | -0.24 | 0.027 | -0.37 | 0.23 |
| Southwestern Finland | 0.08 | 0.0027 | -0.15 | 0.051 |



**Table 4. Observed trends of melting season length during 1982 – 2015 from the CLARA-A2 SAL data of regions based on vegetation zones. The slope and coefficient of determination values are shown both for annual data and moving averages of five years.**

| Region | Annual value | | Moving average of five years | |
|---|---|---|---|---|
| | Slope (Days per year) | $R^2$ | Slope (Days per year) | $R^2$ |
| Northern Fjeld Lapland | -0.63 | 0.097 | -0.62 | 0.22 |
| Forest Lapland | -0.15 | 0.0094 | -0.14 | 0.033 |
| Northwestern Fjeld Lapland | -0.24 | 0.023 | -0.31 | 0.13 |
| North Ostrobothnia | 0.39 | 0.12 | 0.43 | 0.44 |
| Kuusamo district | 0.30 | 0.052 | 0.47 | 0.35 |
| **Southwestern Lapland** | **0.66** | **0.16** | **0.68** | **0.67** |
| Ostrobothnia | 0.20 | 0.022 | 0.33 | 0.18 |
| **Northern Karelia - Kainuu** | **0.58** | **0.12** | **0.71** | **0.68** |
| Lake district | -0.03 | 0.00038 | 0.17 | 0.028 |
| Southwestern Finland | -0.64 | 0.098 | -0.40 | 0.072 |



**Table 5. Observed albedo trends of regions based on vegetation zones. The slope and coefficient of determination values are shown both for annual data and moving averages of five years.**

| Region | Annual value | | Moving average of five years | |
|---|---|---|---|---|
| | Slope (Albedo in % units per year) | $R^2$ | Slope (Albedo in % units per year) | $R^2$ |
| Northern Fjeld Lapland | -0.16 | 0.18 | -0.18 | 0.36 |
| **Forest Lapland** | **-0.23** | **0.52** | **-0.25** | **0.69** |
| Northwestern Fjeld Lapland | 0.03 | 0.01 | -0.02 | 0.01 |
| **North Ostrobothnia** | **-0.34** | **0.72** | **-0.36** | **0.83** |
| **Kuusamo district** | **-0.31** | **0.66** | **-0.31** | **0.82** |
| **Southwestern Lapland** | **-0.33** | **0.61** | **-0.33** | **0.75** |
| Ostrobothnia | -0.14 | 0.23 | -0.08 | 0.47 |
| **Northern Karelia - Kainuu** | **-0.30** | **0.58** | **-0.26** | **0.84** |
| Lake district | -0.03 | 0.006 | 0.06 | 0.15 |
| Southwestern Finland | 0.02 | 0.004 | 0.05 | 0.06 |



**Table 6. Observed albedo trends of regions based on forest centres. The slope and coefficient of determination values are shown both for annual data and moving averages of five years. The region Lapland is divided in northern and southern parts as shown in Figure 1.**

| Region | Annual value | | Moving average of five years | |
|---|---|---|---|---|
| | Slope (Albedo in % units per year) | $R^2$ | Slope (Albedo in % units per year) | $R^2$ |
| **Lapland** | **-0.27** | **0.60** | **-0.30** | **0.72** |
| Northern Lapland | -0.12 | 0.18 | -0.15 | 0.40 |
| **Southern Lapland** | **-0.33** | **0.70** | **-0.34** | **0.82** |
| **North Ostrobothnia** | **-0.21** | **0.48** | **-0.17** | **0.75** |
| Central Finland | -0.005 | 0.0001 | 0.09 | 0.19 |
| Etelä-Savo | -0.05 | 0.01 | 0.04 | 0.06 |
| Pirkanmaa | 0.06 | 0.02 | 0.13 | 0.30 |
| Southwestern Finland | 0.05 | 0.007 | 0.13 | 0.11 |
| Southeastern Finland | -0.02 | 0.002 | 0.06 | 0.08 |
| Häme – Uusimaa | 0.04 | 0.009 | 0.03 | 0.05 |
| **Kainuu** | **-0.30** | **0.58** | **-0.28** | **0.81** |
| South Ostrobothnia | -0.05 | 0.02 | 0.003 | 0.0003 |
| Pohjois-Savo | -0.11 | 0.07 | -0.06 | 0.11 |
| **North Karelia** | **-0.19** | **0.24** | **-0.13** | **0.53** |



Figure 1. The regions based on the vegetation zones (left) and the regions of the forest centres (right) of Finland.

Figure 2. Optimal values of $k$ as a function of the fraction of the forested area in the forest centre area ($f$). The brown point is not included in the regression $k = 0.11 + 1347.44 \exp(-11 f)$, because it represents an area dominated by deciduous species (mountain birch).

Figure 3. The melt onset day of the snow cover in Finland during 1982 – 2011 based on CLARA-A2 SAL product and JSBACH model snow depth.

Figure 4. Trends of the melt onset day of the snow cover in Northern Karelia – Kainuu and Northern Ostrobothnia during 1982 – 2015 based on CLARA-A2 SAL product. The solid curves represent the five year moving averages and the dotted curves the annual values.

Figure 5. Comparison of diverse estimates of the melt-off day of snow cover. The FMI snow maps of end of permanent snow cover are based on in situ snow depth measurements. The satellite based melt-off date estimates are derived using the surface albedo product CLARA-A2 SAL (Anttila et al., 2016, Karlsson et al., 2017) and the FSC snow product calculated from MODIS images (Metsämäki et al., 2018). The JSBACH model estimate of end of snow melt is also shown. For comparison the greening-up date of deciduous vegetation (SYKE DEC; Böttcher et al., 2016) is also presented.

Figure 6. The proceeding of spring in 2006 and 2012 starting from a & e) the melt-off day of the permanent snow cover in Finland based on *in situ* snow depth measurements (Solantie et al., 1996; Kersalo and Pirinen, 2009), b & f) the surface albedo reaching the 20 % threshold value of its dynamic range during the melting season (Anttila et al., 2016, Karlsson et al., 2017), c & g) the complete disappearance of the snow cover based on the FSC product (Metsämäki et al., 2018) and d & h) the greening-up day based on MODIS NDWI (Böttcher et al., 2016).

Figure 7. Trends of the melting season length of the snow cover in Northern Karelia – Kainuu and Northern Ostrobothnia during 1982 – 2015 based on CLARA-A2 SAL product. The solid curves represent the five year moving averages and the dotted curves the annual values.

Figure 8. The surface albedo (%) at the time of the melt onset in Finland. The average value during 1982-1986 (left), during 1982-2015 (middle) and during 2011-2015 (right). The black pixels in Finland represent cases, when the melting has already started before the albedo could be retrieved, i.e. when the sun zenith angle was larger than 60°.

Figure 9. Comparison of observed albedo variation (colour solid curves) of the forest centre areas (Figure 1, Table 2) during 1982 – 2015 and corresponding albedo model results based on stem volume measurements (points). The interpolation curves (grey solid curves) of the modelled annual values are shown as well.





**Figure 1**





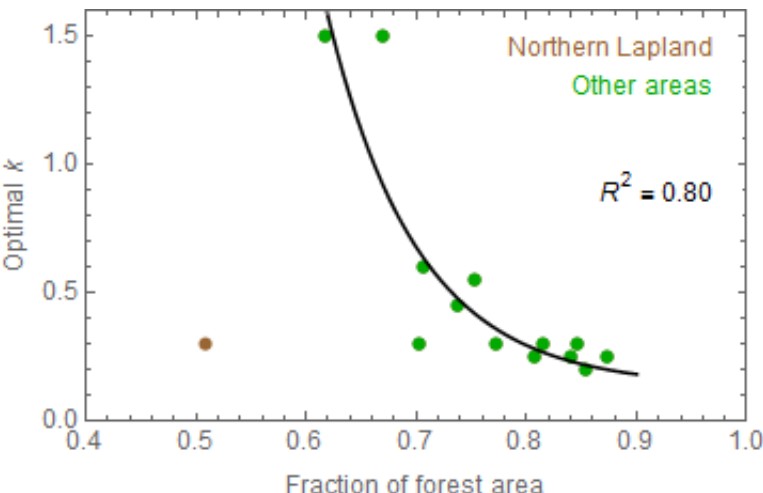

Figure 2




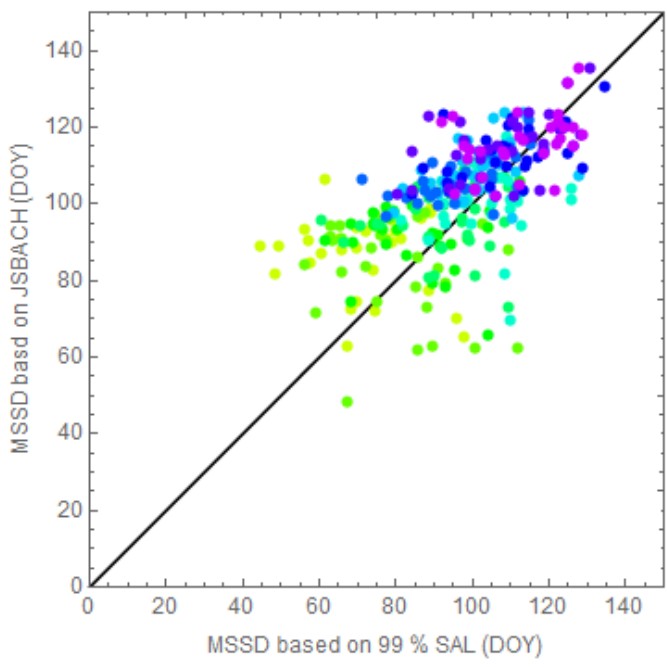

Figure 3





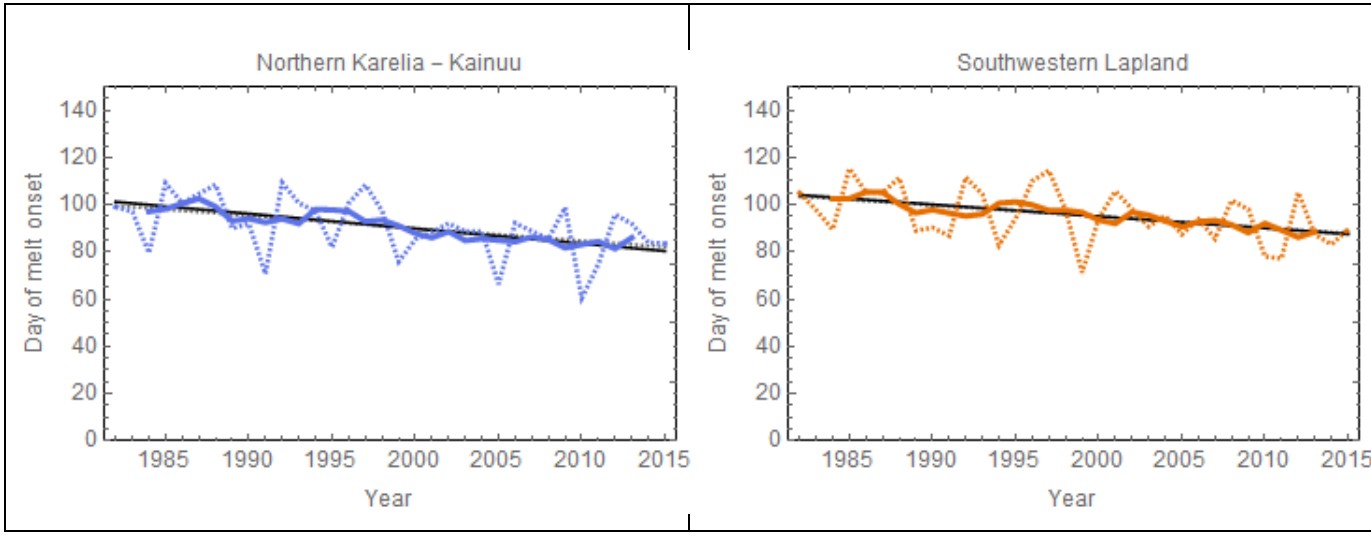

Figure 4





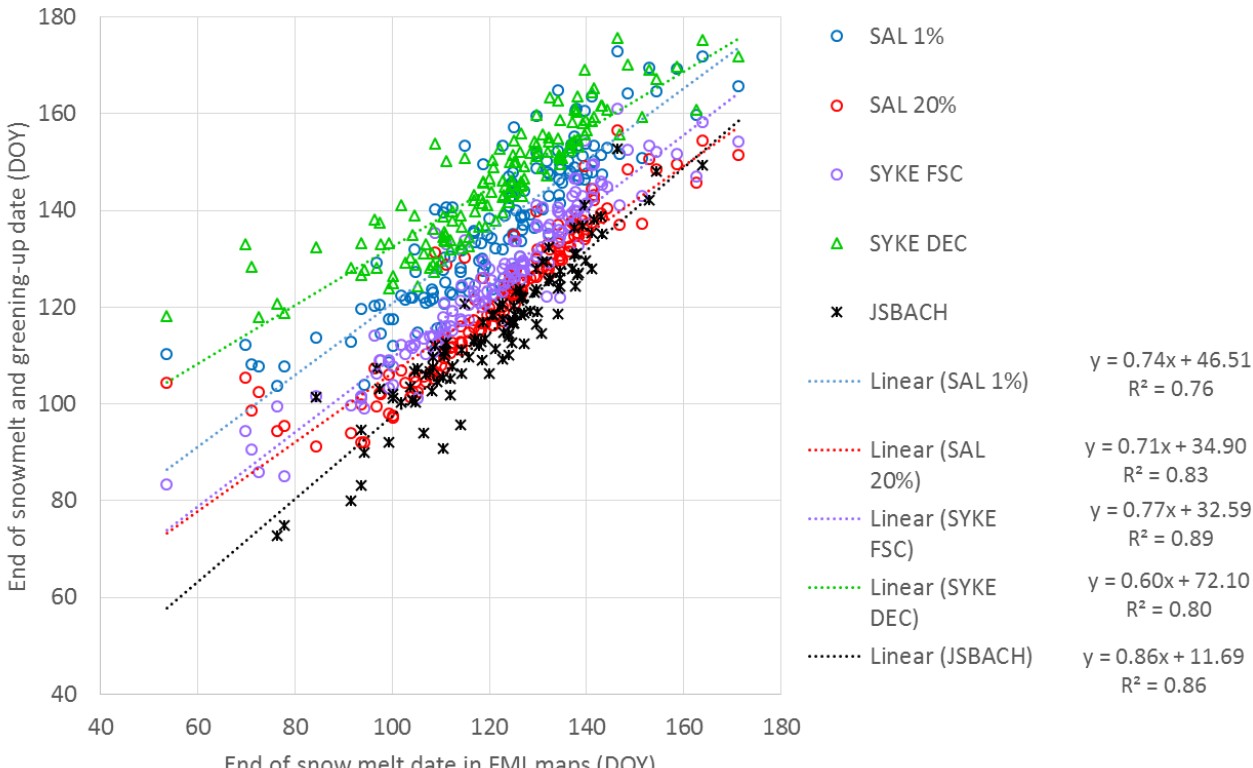

Figure 5





Figure 6





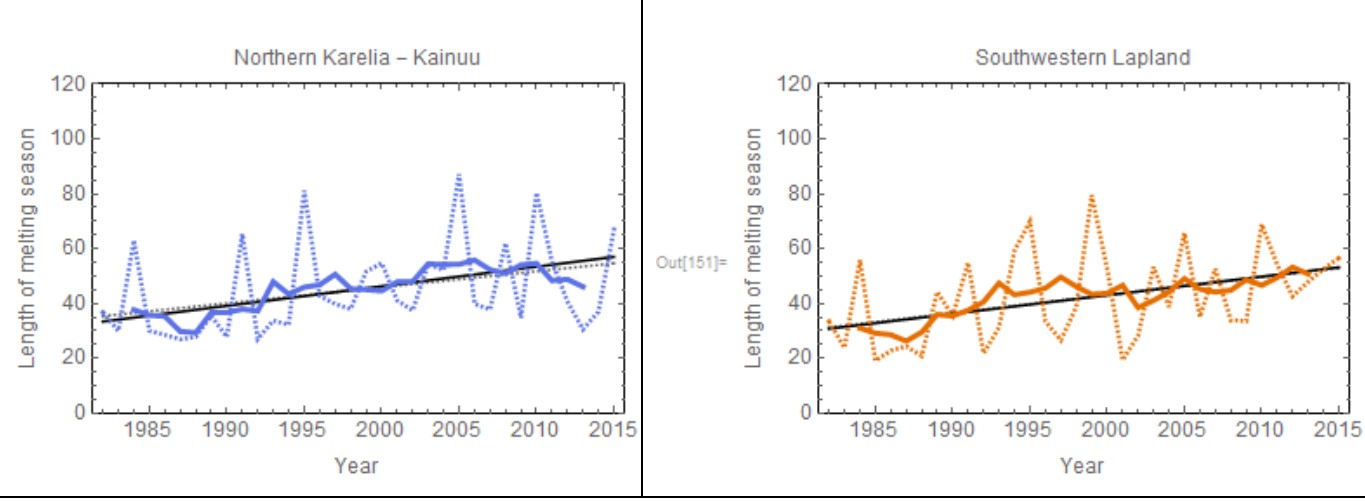

Figure 7



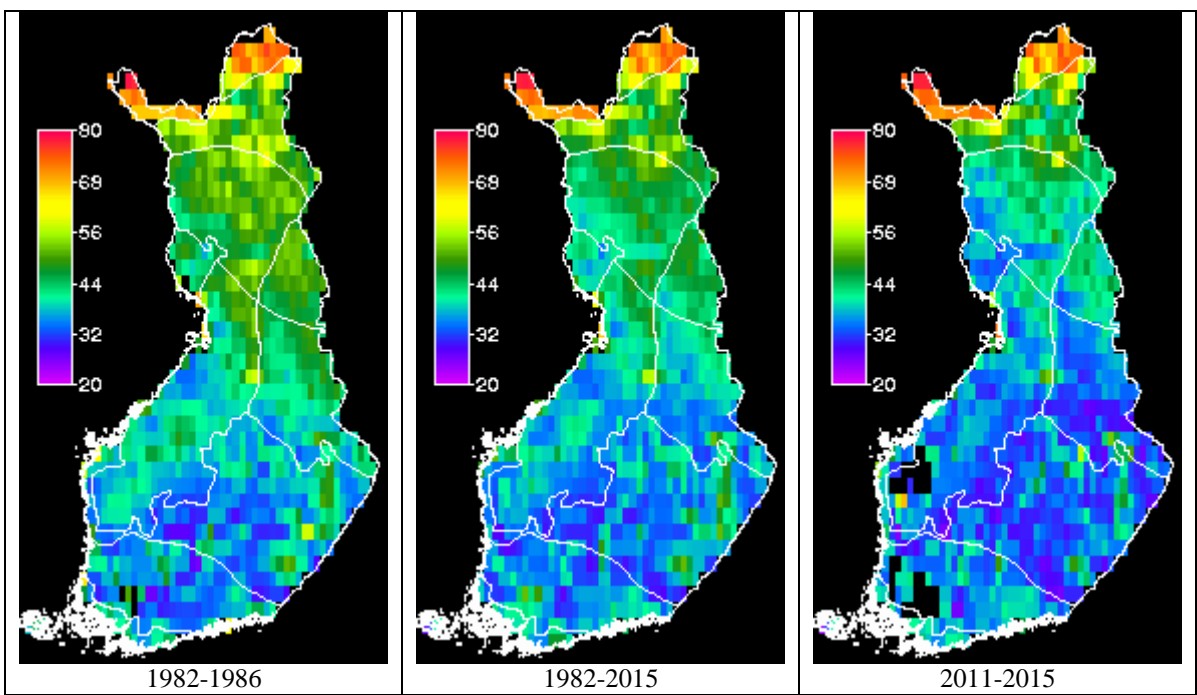

Figure 8





Figure 9