# Peer review of "Monitoring changes in forestry and seasonal snow using surface albedo during 1982–2016 as an indicator"

_Biogeosciences, 2018_

## Referee Comment (RC1) · R. L. H. Essery (Referee) · 19 Sep 2018

This is an interesting and thorough study, but some of the methods are hard to follow. In particular, I recommend some effort on simplifying and clarifying the descriptions of sigmoid fits in sections 2.2.1 and 2.2.2; figures of example fits might help.

Minor comments:

page 1, line 26

Forest cover having a significant influence on albedo does not follow from it being an important component of the carbon budget.

[Figure]

page 2, line 11

"by in" – delete one

page 2, line 24

"since 200" is sufficient

page 3, line 3

I think that standalone forcing of land models with climate model outputs is being discussed here, but it is not clear.

page 4, line 6

Delete "only"

page 5, line 21

"using in-situ FSC observations" Is FSC %-units simply %?

page 6, line 11

What is being iterated here?

page 8, line 3

Roesch et al. (2001) is missing from the reference list

page 9, section 2.2.5

JSBACH should produce an albedo. Has this been examined and rejected for comparison with satellite measurements?

page 9, line 30

Same comment again – can the same variable (albedo) be used for melt onset in data and model?

page 9, line 6

"areal trends"

page 10, line 20

"areas being half"

page 11, section 3.1.3

So is this saying that the snow is starting to melt earlier but is not disappearing any earlier?

page 11, line 29

"as snow or water" (or "as snow or rain")

page 12, line 3

Carelia or Karelia?

page 13, line 11

"also agreed well"

page 13, line 18

"has increased"

Tables 3 to 6

Values in bold font are not explained

Figure 7

Remove Out[151]=

Figure 9

The colours used do not clearly relate to anything

---

## Referee Comment (RC2) · Anonymous Referee #2 · 18 Oct 2018

Manninen et al. use a series of albedo and snow datasets along with the JSBACH land model to evaluate how the characteristics of snowmelt are changing across Finland. They find that some regions exhibit a transition to an earlier start to the melt, but that the melt duration is prolonged. The analysis is very detailed but lacks clarity at some points making it difficult to follow. Although limited in its spatial domain, this paper would be of interest to the community and could be a useful contribution after some issues are addressed.

General comments:

1) My main issue with this manuscript has to do with its readability. There are numerous

grammatical errors throughout, some of which are pointed out below. A native English speaker should edit the paper before resubmission.

2) The authors tend to make some broad sweeping conclusions based on trends that are only significant across a small fraction of the total study region. I would like to see more discussion of the full picture (like Figure 9). Pg 10 L2 and Pg11 L5 are two instances where the discussion is too narrow in focus.

Specific comments:

L25-27: Confusing sentence - reword.

L27: "Change in albedo" to "albedo change". Also, albedo change isn't really a mechanism, but a result of changing forest/snow cover/properties.

L30 and Pg2 L2: "enhances the climate change" – remove the

Pg2 L8: Remove "The forest management changes albedo and carbon sequestration".

L11: remove second "in"

L14: change water to rain

L15: Change to "On average, the precipitation changes..."

L19: Change "snow precipitation" to "snowfall"

L19: "The Arctic warming has resulted in decline of the Arctic Ocean ice cover" Awkward wording, change to this or similar: Arctic warming has caused Arctic sea ice cover to decline rapidly.

L20: Change "such decline" to "such a decline"

L24: since 2005

L26: Based on the prior paragraphs, the authors should mention how well models represent recent changes in snow cover (e.g., Derksen and Brown, 2012, GRL; Thackeray
et al., 2016, J Climate).

Pg3 L23-26: This sentence should be reworded.

Pg4 L5: Change "the spatial resolution" to "a spatial resolution".

L10: remove second "of Finland" here.

L11: change "forest associated variables"

L17-19: Confusing sentence, reword.

Pg5 L19: Change to "if the forest floor"

L21: Confused by "15-20 FSC %-units".

L23: Confusing first sentence, reword.

Pg7 L17-23: This paragraph seems as though it would be a better introduction to this section.

Pg8 L4: Change "composes" to "is composed"

L22: This sentence is poorly worded. Do the authors mean to say that Snow melt-off day is derived from FSC maps?

L23: Change "however ignoring" to "but ignores"

Pg9 L25: Change to "Seasonal snowmelt timing" or similar?

Pg10 L1: Since these changes are so small across many regions, I recommend changing the units from days per year to days per decade (Table 3,4,5, etc).

L2: What about the other eight vegetation zones?

L20: Change "The land ecosystem model results" to "Results from JSBACH"

L21-22: Awkward sentence, reword.

L33: Not clear what this means: "the southeastern -northwestern zones of equal timing are obvious".

Pg11 L5: What about the two regions with a decreasing melting period length of a similar magnitude?

L6: Change to "Start of the Growing Season"

L9: should be "on the forest floor"

L27: Remove "the" to start the sentence and change "quality" to "type".

L29: Remove "as form of snow or water"

L29: Capitalize arctic

Combine Tables 3, 4 and 5 into one to save space?

Table 3-5: state in the caption why some entries are bold.

Figure 1, 6, 8: Use white background with black outlines. Include latitude/longitude gridlines.

Figure 7: remove "out[161]=" from plot

Figure 8: Include a panel showing the explicit albedo change from the 1982-1986 panel to the 2011-2015 panel. This will make it easier for the reader to comprehend the change.

Figure 9: Narrow the y-axis on these plots to make the interannual variability and change more apparent (10 or 20 to 60%?).

---

## Author Comment (AC1) · 14 Nov 2018

**Answers to the comments by reviewer RC1:**

This is an interesting and thorough study, but some of the methods are hard to follow. In particular, I recommend some effort on simplifying and clarifying the descriptions of sigmoid fits in sections 2.2.1 and 2.2.2; figures of example fits might help.
The authors are grateful for the encouraging comment. The sigmoid fitting is described now in more detail and related images are included. The method sounds complicated, because one has to take into account also rare, but possible complications (such as marked snowfall after melt onset). However, the main complication really to take into account is the greening up of vegetation. This is demonstrated in the added Figure 2.

[Figure]

Figure 2. Examples of sigmoid fitting. Original data points are shown in blue, the fit in red and the data points from which vegetation greening effect has been removed in black. a) initial fit of an easy case, b) final fit of the easy case, c) initial fit of a case with strong influence of vegetation greening on albedo, d) final fit of the case with strong influence of vegetation greening on albedo.

Minor comments:

page 1, line 26.
Forest cover having a significant influence on albedo does not follow from it being an important component of the carbon budget.
This sentence is now edited according to the comment.

page 2, line 11.
"by in" – delete one
"in" is deleted as requested.

page 2, line 24
"since 200" is sufficient
"Year" is removed as suggested.

page 3, line 3
I think that standalone forcing of land models with climate model outputs is being discussed here, but it is not clear.
The text is edited (page 3, line 6).

page 4, line 6
Delete "only"
"Only" is deleted as requested.

page 5, line 21
"using in-situ FSC observations" Is FSC %-units simply %?
Yes, the FSC units is %. There was an extra confusing FSC that is now removed. The FSC unit is defined to be in % on page 5, line 21.

page 6, line 11
What is being iterated here?
The sigmoid fitting is iterated means that the nonlinear fitting is made iteratively as usual. The reason for this is that the nonlinear fit result depends much on the initial values of the parameters of the function to be fitted. As there is no absolute method to find the best possible initial values, it is useful to iterate the results so that the next fit uses as initial parameter values the outcome of the previous iteration. This section (page 6, lines 12 – 26) is now described in more detail as suggested and Figure 2 is added.

page 8, line 3
Roesch et al. (2001) is missing from the reference list
The missing reference has been added.

page 9, section 2.2.5
JSBACH should produce an albedo. Has this been examined and rejected for comparison with satellite measurements?

Yes, the albedo was also compared and the results were the same as using the snow depth. Here we chose the snow depth, because the model provides it and our main 'ground truth' was the operational snow depth. In addition, snow depth that accumulates throughout the seasonal cold period is a better indicator of winter-time climate than snow albedo, which is sensitive to prevailing weather conditions before the melt onset. Hence the scatter between the model and satellite data based albedo is larger than that of the end of melt season. Below the comparison between the albedo values at the beginning and end of snow melt are shown.

[Figure]

page 9, line 30
Same comment again – can the same variable (albedo) be used for melt onset in data and model?

Yes, the albedo of the model can be used also for the melt onset observation. The scatter is slightly larger in that case than for the end of snow melt, because the true albedo right before the melt onset is very sensitive to prevailing weather conditions. Related text is added on page 10, lines 9-11.

page 9, line 6
"areal trends"
Now page 10, line 18. Edited as requested.

page 10, line 20
"areas being half"
This sentence is reworded (page 11, line 2).

page 11, section 3.1.3
So is this saying that the snow is starting to melt earlier but is not disappearing any earlier?
Yes, that's right. The end of melt season is very sensitive to the weather conditions at that time and has no clear trend, whereas the onset of melt is related to the weather conditions during a longer time period and has a decreasing trend.

page 11, line 29
"as snow or water" (or "as snow or rain")
Edited as suggested.

page 12, line 3
Carelia or Karelia?
The authors use the official English forms of the names, however odd they seem. Hence, Carelia but Kainuu. Häme – Uusimaa (in Finnish), but Ostrobothnia (Latin version of Etelä-Pohjanmaa)!

page 13, line 11
"also agreed well"
Edited as suggested.

page 13, line 18
"has increased"
Edited accordingly.

Tables 3 to 6
Values in bold font are not explained
The idea was to emphasize the cases for which the moving average of five years shows a significant coefficient of determination (>0.5). This information is now added to the table captions.

Figure 7
Remove Out[151]=

Removed.

Figure 9
The colours used do not clearly relate to anything
The colours are related to the map of Figure 1b. This information is added to the Figure caption 10. As the whole Lapland is treated as one, the colour is a random choice.

One co-author from the FMI model team should be added:
Antti Leppänen, Finnish Meteorological Institute

---

## Author Comment (AC2) · 14 Nov 2018

**Answers to the comments by reviewer RC2:**

General comments:
1) My main issue with this manuscript has to do with its readability. There are numerous grammatical errors throughout, some of which are pointed out below. A native English speaker should edit the paper before resubmission.
This is now done. The authors are grateful for the thorough review also concerning the language.

2) The authors tend to make some broad sweeping conclusions based on trends that are only significant across a small fraction of the total study region. I would like to see more discussion of the full picture (like Figure 9). Pg 10 L2 and Pg11 L5 are two instances where the discussion is too narrow in focus.
The authors have now extended the mentioned discussions (sections 3.1.1, 3.1.3 and 3.3.) as requested. The systematic signal of change is not as dramatic in Finland as in some areas, because of the large variation of winter weather. Especially the sea ice extent of Baltic Sea affects largely the Finnish winter weather. If the Gulf of Finland and Gulf of Bothnia freeze completely the Finnish climate is almost continental in winter even on the coast. We added a comment about the hemispherical study published recently, where we studied the effect of weather parameters on the observed changes in larger areas.

Section 3.1.1 main addition:
Especially variable melt onset timing is in the coastal regions (Southwestern Finland and Southern Ostrobothnia) and in the Lake district. For those regions the standard deviation values of the melt onset day are 14.3, 14.7 and 14.6, respectively. For the Northern Fjeld Lapland the vicinity of the Barents Sea obviously causes also higher a standard deviation (13.4) of the melt onset day. So large part of Finland is coastal that changes in air temperature or precipitation preceding melt onset are not that directly dominating as in some regions of the northern hemisphere (Anttila et al., 2018), but the sea ice extent of the Baltic Sea has a large effect as well. When using 10 year moving averages for the trend analysis based on albedo data, a negative trend ($R^2 > 0.5$) of melt onset day was detected in Northern Karelia – Kainuu, Southwestern Lapland, Ostrobothnia, Kuusamo district and Lake district. Although the time series of 34 years is not really long enough for using 10 year averages, the results, however, confirm the intuitive impression that the snow melt starts earlier than it used to do in the past, as the two distinct areal trends showed.

Section 3.1.3 addition:
For the length of the melting season the standard deviation during 1982-2015 was on the average, naturally, even larger than for the melt onset day the value being as high as 16 days. The largest values occurred again in Southwestern Finland (20.3), Southern Ostrobothnia (17.5), Lake district. (17.1) and Northern Fjeld Lapland (20.0). When using 10 year moving averages in the trend analysis, the result was that the melting season length increased in Northern Karelia – Kainuu, Kuusamo district, Southwestern Lapland and North Ostrobothnia and decreased in Northern Fjeld Lapland.

Section 3.3. addition:
It is obvious also that meaningful trends ($R^2 > 0.5$) for five year average albedo values were obtained only when the fraction of forested area exceeded 0.7 and the stem volume was smaller than 75 m$^3$/ha. For those areas (Southern Lapland, Kainuu, Northern Ostrobothnia, Lapland and North Karelia) the albedo trend is negative. It is well known that the winter time albedo depends strongly on the LAI (and thus stem volume of a larger area), when the LAI is relatively small (Manninen and Stenberg, 2009). For larger LAI values the albedo more or less saturates and a further increase of LAI does not show up markedly.

Specific comments:
L25-27: Confusing sentence - reword.
Edited as requested.

L27: "Change in albedo" to "albedo change". Also, albedo change isn't really a mechanism, but a result of changing forest/snow cover/properties.
Edited as requested.

L30 and Pg2 L2: "enhances the climate change" – remove the
Edited as requested.

Pg2 L8: Remove "The forest management changes albedo and carbon sequestration".
Edited as requested.

L11: remove second "in"
Edited as requested.

L14: change water to rain
Edited as requested.

L15: Change to "On average, the precipitation changes..."
Edited as requested.

L19: Change "snow precipitation" to "snowfall"
Edited as requested.

L19: "The Arctic warming has resulted in decline of the Arctic Ocean ice cover" Awkward wording, change to this or similar: Arctic warming has caused Arctic sea ice cover to decline rapidly.
Edited as requested.

L20: Change "such decline" to "such a decline"
Edited as requested.

L24: since 2005
Edited as requested.

L26: Based on the prior paragraphs, the authors should mention how well models represent recent changes in snow cover (e.g., Derksen and Brown, 2012, GRL; Thackeray et al., 2016, J Climate).
These mentioned essential references are now added, thanks for suggesting.

Pg3 L23-26: This sentence should be reworded.
This sentence is split in two to clarify the text.

Pg4 L5: Change "the spatial resolution" to "a spatial resolution".
Edited as requested.

L10: remove second "of Finland" here.
Edited as requested, although now it is implicitly expected to be understood that they don't survey forests abroad.

L11: change "forest associated variables"
Edited.

L17-19: Confusing sentence, reword.
Edited.

Pg5 L19: Change to "if the forest floor"
Edited as requested.

L21: Confused by "15-20 FSC %-units".
They are just percentages. Edited accordingly.

L23: Confusing first sentence, reword.
The authors did not grasp what is confusing, but tried to clarify this sentence.

Pg7 L17-23: This paragraph seems as though it would be a better introduction to this
section.
The paragraph is moved to the beginning of section 2.2.2.

Pg8 L4: Change "composes" to "is composed"
Edited as requested.

L22: This sentence is poorly worded. Do the authors mean to say that Snow melt-off
day is derived from FSC maps?
Yes. The first sentence is now removed altogether as the later text explains, how the snow melt-off day is derived.

L23: Change "however ignoring" to "but ignores"
Edited as requested.

Pg9 L25: Change to "Seasonal snowmelt timing" or similar?
Edited as requested.

Pg10 L1: Since these changes are so small across many regions, I recommend changing
the units from days per year to days per decade (Table 3,4,5, etc).
Tables 3, 4, 5 and 6 are edited as requested. The corresponding change has been made also in the text part, where
numbers appear (abstract and conclusions).

L2: What about the other eight vegetation zones?
The other vegetation zones did not show distinct trends. This information is added in the text on page 10, lines 14-15.

L20: Change "The land ecosystem model results" to "Results from JSBACH"
Edited as requested.

L21-22: Awkward sentence, reword.
Edited as requested.

L33: Not clear what this means: "the southeastern -northwestern zones of equal timing
are obvious".
The snow melt contours in Finland are more or less in the southeast-northwest direction. The sentence is edited (page
11, line 16).

Pg11 L5: What about the two regions with a decreasing melting period length of a
similar magnitude?
The coefficients of determination for those areas are markedly smaller than 0.5, so that the value of the slope does not
signify, not even its sign.

L6: Change to "Start of the Growing Season"
Edited as requested. Also elsewhere in the manuscript we now use the term start of the growing season.

L9: should be "on the forest floor"
Edited as requested.

L27: Remove "the" to start the sentence and change "quality" to "type".
Edited as requested.

L29: Remove "as form of snow or water"
Edited as requested.

L29: Capitalize arctic
Edited as requested.

Combine Tables 3, 4 and 5 into one to save space?
The tables 3 and 4 are combined, because the significant trends of melt onset and melting season length appear in the same regions (shown in bold fonts). Table 5 (now 4) is kept separate, because the albedo trends are significant in other regions than those of the melt onset and melting season length. Besides, table 5 (now 4) will be compared more with table 6 (now 5).

Table 3-5: state in the caption why some entries are bold.
Edited as requested.

Figure 1, 6, 8: Use white background with black outlines. Include latitude/longitude gridlines.
The background and outlines are updated as requested. The region maps are given in the national metric coordinate system, not in latitudes/longitudes. In order to avoid conversion errors we carried out the analysis in the original coordinates. The latitude/longitude information is now added to the caption of Figure 1. As all figures cover the same area, it is sufficient to give it once.

Figure 7: remove "out[161]=" from plot
Edited as requested.

Figure 8: Include a panel showing the explicit albedo change from the 1982-1986 panel to the 2011-2015 panel. This will make it easier for the reader to comprehend the change.
The authors agree that this is a good idea and added the requested panel.

Figure 9: Narrow the y-axis on these plots to make the interannual variability and change more apparent (10 or 20 to 60%?).
Updated as requested.

One co-author from the FMI model team should be added:
Antti Leppänen, Finnish Meteorological Institute

---

## Author Response (AR2)

Answers to the editor:

The authors have now read the manuscript anew and tried to check the English for the final version and to improve the readability. Basic checking of all images, tables, references and acknowledgements was also carried out anew. The co-author that we got the permission to add was erroneously missing from the previously submitted version.

Specific answers:

Line 9: The surface albedo time series, CLARA-A2 SAL, was used

The commas are now added.
Line 11-12: In addition, the melt onset from JSBACH was compared with the timing of green-up estimated from MODIS data.

Edited as requested.
Line 12: Similarly, the end of snow melt predicted by JSBACH was compared with ...

Edited as requested.
Line 14-onwards: "It was found that 15 the albedo threshold 20 % of the ..." makes no sense.

This comment puzzled the authors. The number 15 is the number of the line in question, not part of the sentence. This is completely clear from the manuscript provided by the link uploaded by Anna Wenzel (27 Nov 2018). Are the reviewers/editor reading some other, further processed version of the manuscript? The authors can check only the versions available for them.

Line 16: See above.

The meaning of this comment is difficult to guess, as the previous comment was not related to the contents of the manuscript. The authors tried to clarify the text.

Line 17: In two northern vegetation map areas (vague), a clear trend towards an earlier snow melt onset ...

The comma is now added and the areas in question are specified.

Line 20: "The increased stem volume explained the trend." - this is not a complete sentence, you need further details.

The authors thought the sentence would be understandable from the context, but anyhow completed the sentence.